# Process Transferability of Friction Riveting of AA2024-T351/Polyetherimide (PEI) Joints Using Hand-Driven, Low-Cost Drilling Equipment

Anamaria Feier [1] , Andrei Becheru [1], Mihai Brînduşoiu [1] and Lucian Blaga [2,*]

1 Department of Materials and Manufacturing Engineering, Mechanical Faculty, Politehnica University Timişoara, Bl. Mihai Viteazu No. 1, 300222 Timişoara, Romania; ana-maria.feier@upt.ro (A.F.); becheru.andrei@yahoo.ro (A.B.); mihaibrindusoiu@gmail.com (M.B.)
2 Department Solid State Materials Processing, Institute for Materials Mechanics, Helmholtz Zentrum Hereon, 21502 Geesthacht, Germany
* Correspondence: lucian.blaga@hereon.de; Tel.: +49-(0)4152-87-2055

**Abstract:** The present work deals with the transferability of Friction Riveting joining technology from laboratory equipment to adapted in-house, low-cost machinery. A G13 drilling machine was modified for the requirements of the selected joining technique, and joints were performed using polyethermide plates and AA2024 aluminum alloy rivets of 6 mm diameter. This diameter was not previously reported for Friction Riveting. The produced joints were mechanically tested under tensile loading (pullout tests) with ultimate tensile forces of $9500 \pm 900$ N. All tested specimens failed through full-rivet pullout, which is the weakest reported joint in Friction Riveting. In order to understand this behavior, FE models were created and analyzed. The models produced were in agreement with the experimental results, with failure initiated within the polymer under stress concentrations in the polymeric material above the deformed metallic anchor at an ultimate value of the stress of 878 MPa at the surface of the joint. Stresses decreased to less than half of the maximum value around the anchoring zone while the rivet was removed and towards the surface. The paper thus demonstrates the potential ease of applying and reproducing Friction Riveting with simple machinery, while contributing to an understanding of the mechanical behavior (initialization of failure) of joints.

**Keywords:** Friction Riveting; metal-polymer hybrid joints; friction-based multi-material connections; anchoring FE modelling; rivet failure modes

## 1. Introduction

The continuously increasing demand for cost reduction, together with high performance product requirements, is leading to substantial research and engineering developments in new materials and tailored joining technologies [1]. The outcomes of these efforts are mixed and hybrid structures in which the properties and performance of products are improved through combining the properties and behaviors of each specific material [2]. Hybrid polymer–metal structures are used in such way in a several engineering applications, as exemplified in Figure 1.

Due to strong dissimilarities in physical-chemical properties, hybrid joints between metals and polymers are challenging, more so because of geometrical and design considerations [2]. To overcome some of the limitations of the current state of the art in hybrid joining, Friction Riveting (FricRiveting) has been developed at the Helmholtz-Zentrum Geesthacht (now Helmholtz Zentrum Hereon) in Germany as a process for joining metallic bolts (rivets) with polymeric plates [3]. This research studies the material combination of aluminum AA204 with polyetherimide (PEI) to be joined via Friction Riveting. The feasibility of this combination has already been proven, and the joints have been characterized and

optimized in a series of studies using the equipment available at the institute that holds the patent for Friction Riveting [4–6].

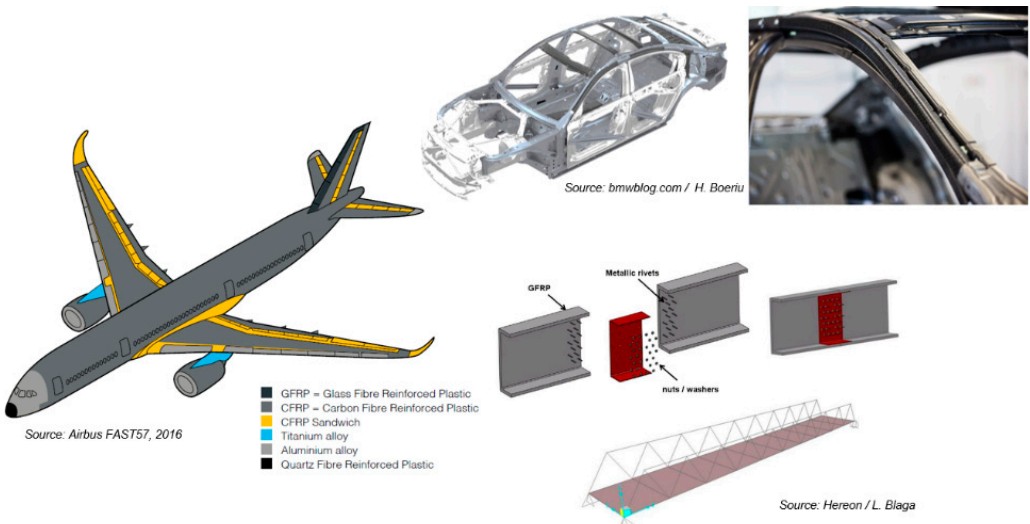

**Figure 1.** Application examples of hybrid polymer–metal joints.

Friction Riveted metallic-insert/point-on-plate joints for polymer–metal hybrid structures could be used in transportation industries or civil engineering, as well as in lower-scale electronics, in clips, stinger-skin connections, sandwich panels, or additional safety connections to welded or adhesively bonded structures. Such joints can be applied both to axial loads, whereby the rivet anchoring efficiency is of uttermost importance, and to shear loads, in which the joint configuration is single-, or double-lapped [2]. The scope of this research paper is to validate the previous experiments for Friction Riveting under axial loading (anchoring efficiency) by using and partially adapting an existing drilling machine, contributing thus to the industrial transferability of the technique. Furthermore, the paper intends to generate new knowledge regarding the failure mode and fracture initiation within the polymeric part resisting the removal of the anchored rivet.

## 2. Friction Riveting

In its basic process variant involving the so-called point-on-plate or metallic-insert joint (Figure 2), Friction Riveting consists of a rotating cylindrical metallic rivet being plastically deformed and subsequently anchored within a polymer or polymeric composite plate. The joining occurs mainly through the mechanical interlocking and anchoring of the plastically deformed tip of the metallic rivet inside the polymer part. The rotating rivet widens its diameter inside the polymer because of the increasing heat due to the friction, corroborated by heat concentration in the joint formation area due to the insulating properties of polymers [7]. Adhesion between the partially softened or molten polymer after its reconsolidation might also contribute to the joining mechanisms, as well as to the micromechanical interlocking at the rivet–polymer interface. The softened/molten polymeric material from the joining area, along the path of the inserting rivet, is expelled during the process as flash material, which can be removed during or after the process by cutting [8].

Despite significant studies and characterizations of different material combinations in Friction Riveting, the technique was not applied outside of laboratory environments and industrial machinery. The scope of this research paper is to validate the previous experiments by using and partially adapting an existing drilling machine, contributing thus to the industrial transferability of the technique and concomitantly to the modelling of joint mechanical behavior for an improved understanding of the failure of such joints under axial loading.

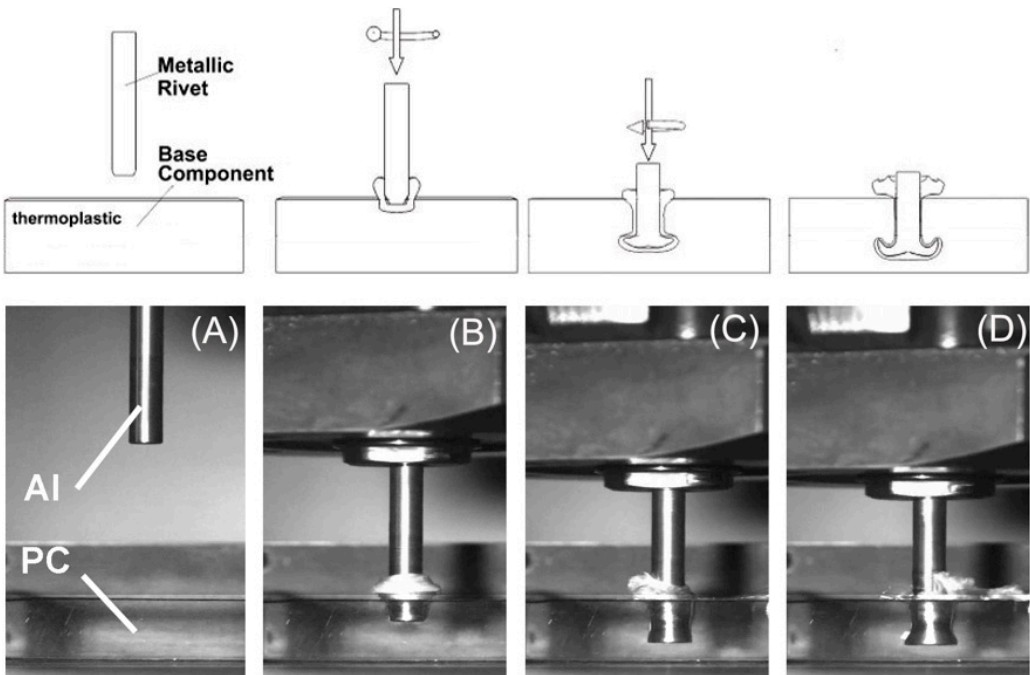

**Figure 2.** Friction Riveting process steps shown on an aluminum–polycarbonate joint: (**A**) Clamping/positioning of joining partners, (**B**) Rivet rotation and insertion, (**C**) Rivet plastic deformation and forging, (**D**) Joint consolidation/anchoring [7]. Reproduced with permission from [7].

## 3. Materials and Methods

### 3.1. Base Materials

Aluminum AA2024-T351 extruded rivets were joined with polyetherimide (PEI) plates, the exact material combination and specifications as in [5,6]. Plain, featureless rivets were used, with a diameter of 6 mm and 60 mm length. The choice of materials was based on the process development of Friction Riveting, as this was the initial material combination used for the process development and validation [2].

PEI is a high-performance thermoplastic amorphous polymer characterized by high specific strength, increased rigidity under high temperatures, thermal stability, and good electrical properties. It is used widely in electronics and medical devices as well as the chemical and petroleum industries [9]. The AA2024-T3521 alloy is widely used in the transportation industries due to its relatively high strength under both quasi-static and dynamic loading, while its corrosion resistance is rather moderate.

### 3.2. Joining Equipment and Procedure

The equipment used in this study was a G13 drilling machine (producer, country), presented in Figure 3. The G13 is a fixed-tool drilling machine for the small-series drilling of steel up to Ø 13 mm, weighing a total of 162 kg.

As the maximum rotational speed of the G13 is 4000 rpm, it was by far not matching the equipment used for previous work in FricRiveting, where rotational speeds of up to 23,400 rpm were applied to join the same combination of materials, with a reduced rivet diameter indeed, of only 5 mm. Therefore, the equipment was modified by replacing the original motor with a commercial, off-the-shelf motor capable of achieving 10,000 rpm rotational speed. A hand-driven force of 1950 N was applied, constant for all trials. The displacement of the rivet (penetration rate) was manually controlled via an indicator on the drilling machine. The successfully anchored joints tested and reproduced were performed using the maximum configuration of this adapted equipment, with any lower setting not capable of obtaining sufficient deformation and anchoring.

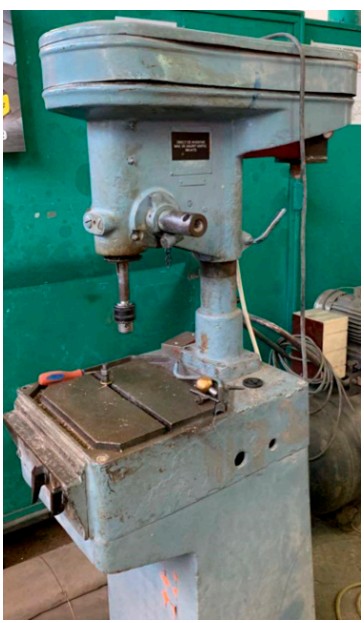

**Figure 3.** G13 drilling machine used for Friction Riveting.

### 3.3. Joint Mechanical Performance

Five specimens were produced via the described method, and their mechanical performance was tested in terms of ultimate pull-out force using a Testwell/UTS universal testing machine with a 260 kN load cell. Tests were performed at room temperature at a testing speed of 1 mm/min. The testing configuration is shown in Figure 4.

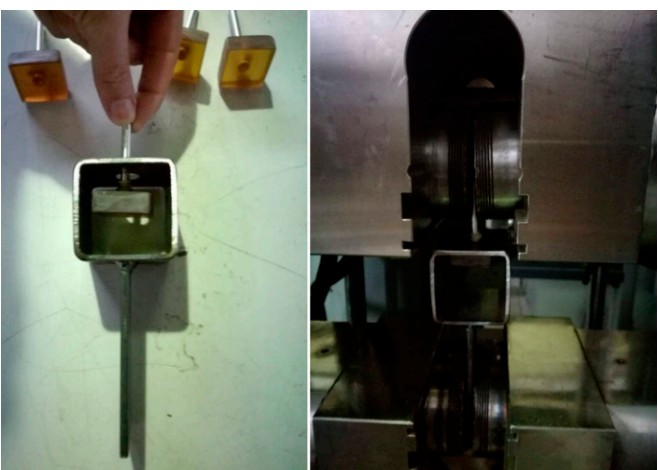

**Figure 4.** Pullout tests of friction-riveted joints.

### 3.4. Numerical Simulation of Pullout Tests

The joints were modelled numerically via ANSYS v19.1 FEM software. First, the joining partners were discretized and the material properties defined. Figure 5 shows the cross section of a joint used for FE modelling in this study.

For the discretization of the model, tetrahedral elements were applied in the meshing. In the anchoring zone/joining area, a coarse mesh was used for an improved evaluation of the results. The mechanical properties in Table 1 were used as material inputs for the FE model.

**Table 1.** Mechanical properties of materials used in FE model.

| Component | Material | Density (g/cm$^3$) | E Modulus (MPa) | $\nu$ (-) | $R_m$ (MPa) | $R_{p0.2}$ (MPa) |
|---|---|---|---|---|---|---|
| Rivet | AA 2024-T351 | 2.78 | 76,000 | 0.33 | 470 | 324 |
| Base plate | PEI | 1.27 | 3500 | 0.44 | 129 | 129 |

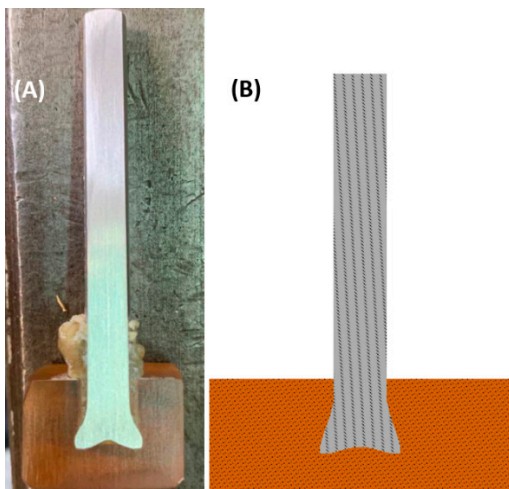

**Figure 5.** Friction riveted joint (**A**) cross section and (**B**) FE model.

## 4. Results and Discussion

As described in Section 3.2, AA2024-T351 rivets were successfully deformed and anchored within PEI plates. The final joining parameters were a rotational speed of 10,000 rpm, a joining force of 1950 N, and displacement at friction manually controlled as described in Section 3.2. Lower rotational speeds were tested but led to insufficient deformation and thus no presence of rivet anchoring, while increased loads produced either rivet buckling or full drilling of the PEI plate.

Four specimens were mechanically tested, as described in Section 3.3, leading to an ultimate pullout force of 9500 ± 900 N. All tested samples failed by full rivet pullout, as shown in Figure 6.

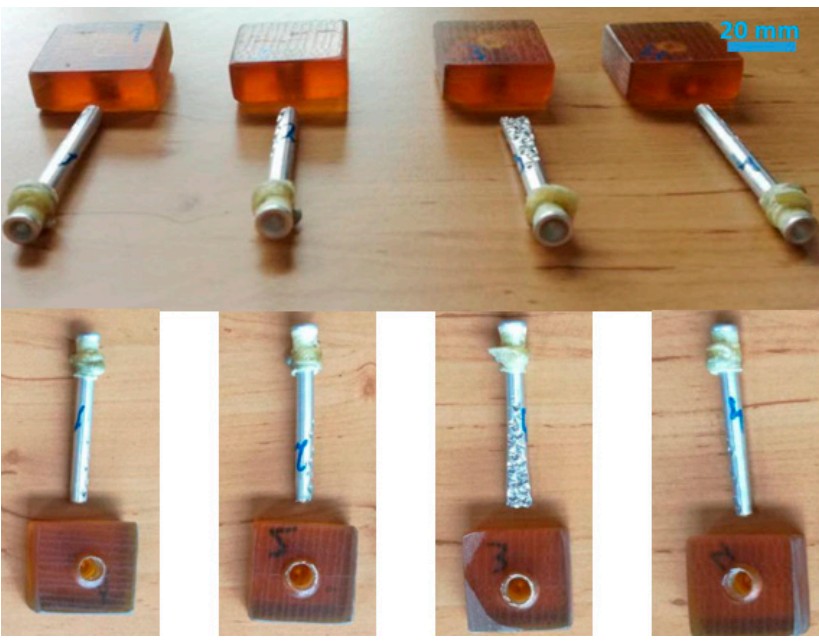

**Figure 6.** Tested specimens displaying full rivet pullout failure.

One can observe that a certain volume of polymeric material is adhered to the rivet. This consists of both the flash expelled during the friction riveting process, as well as a narrow layer of polymer displaced from the composite plate's surface, which corresponds to the full rivet pullout failure observed by Pina Cipriano et al. [6] and first identified as Type III failure of Friction Riveting joints by Rodriguez et al. [7].

For a better comprehension of the mechanical behavior of the anchored rivet under tensile loading, the results of the mechanical tests were compared with the FE models. Figure 7 presents a numerical model of the development of the deformations and Misses stresses during the rivet displacement.

The ultimate failure (Figure 8) occurs in the experiments via complete removal of the anchored rivet, without additional cracks in the polymeric material. The ultimate value of the stress was 878 MPa, present only at the surface of the joint, indicating the weakening of the polymeric material above the deformed rivet, leading to a lower resistance to the removal of the metallic anchor. Deeper in the joining area, close to the actual anchoring, the stresses decrease to less than a half of the surface value; therefore, the rivet does not achieve the ultimate yield stress and is instead removed and shifts toward the surface. Furthermore, the initialization of the removal of the rivet from the anchoring area can be observed. Figure 9 compares the stress–stress curves of both the FE model and the experimental data, showing their agreement. Pina Cipriano et al. previously investigated the correlation of the anchoring deformation, ultimate tensile force, and subsequent joint failure for the same material combination as in the present paper [5,6]. By addressing the anchoring efficiency in terms of volumetric ratio and via statistical analysis, they showed that the most significant process parameter is the joining force (friction force in their work). It has to be noted that the work of Pina Cipriano et al. was performed on dedicated laboratory equipment, specifically designed for Friction Riveting [5,6]. As also mentioned in their research report, the joining force is of highest importance for joint formation, given sufficient heat input provided by the parameters of rotational speed and the rotational speed–joining force interaction. The current study shows to some extent that the rotational speed required can be as low as 10,000 rpm for deformation and anchoring to occur. Indeed, increased mechanical performance and an improved failure type still has to be accomplished when transferring the technology to simpler devices. The recommendation would be to increase the joining force and future research will have to confirm this. Nonetheless, factors such as rivet buckling and oversized deformation (leading to weak anchoring) will have to be considered in that case, as well as joining-energy efficiency.

Borges et al. analyzed previous FE models of PEI/AA2024 Friction Riveting joints [10], which cannot be applied directly to the current work, as the joining conditions, equipment, and most important, the mechanical behavior differ. In the work of Borges et al., optimized joints were modelled with a final fracture within the shaft of the metallic rivet. The disagreement between models and experiments of around 10% was supposedly due to geometrical simplifications [10]. Moreover, the Johnson–Cook failure model was applied to the rivet, which could not be the case in the present investigation, as the failure initiates and finalizes within the polymeric material only (metallic rivet completely removed with some adhesion and fracture at the PEI-AA2024 interface).

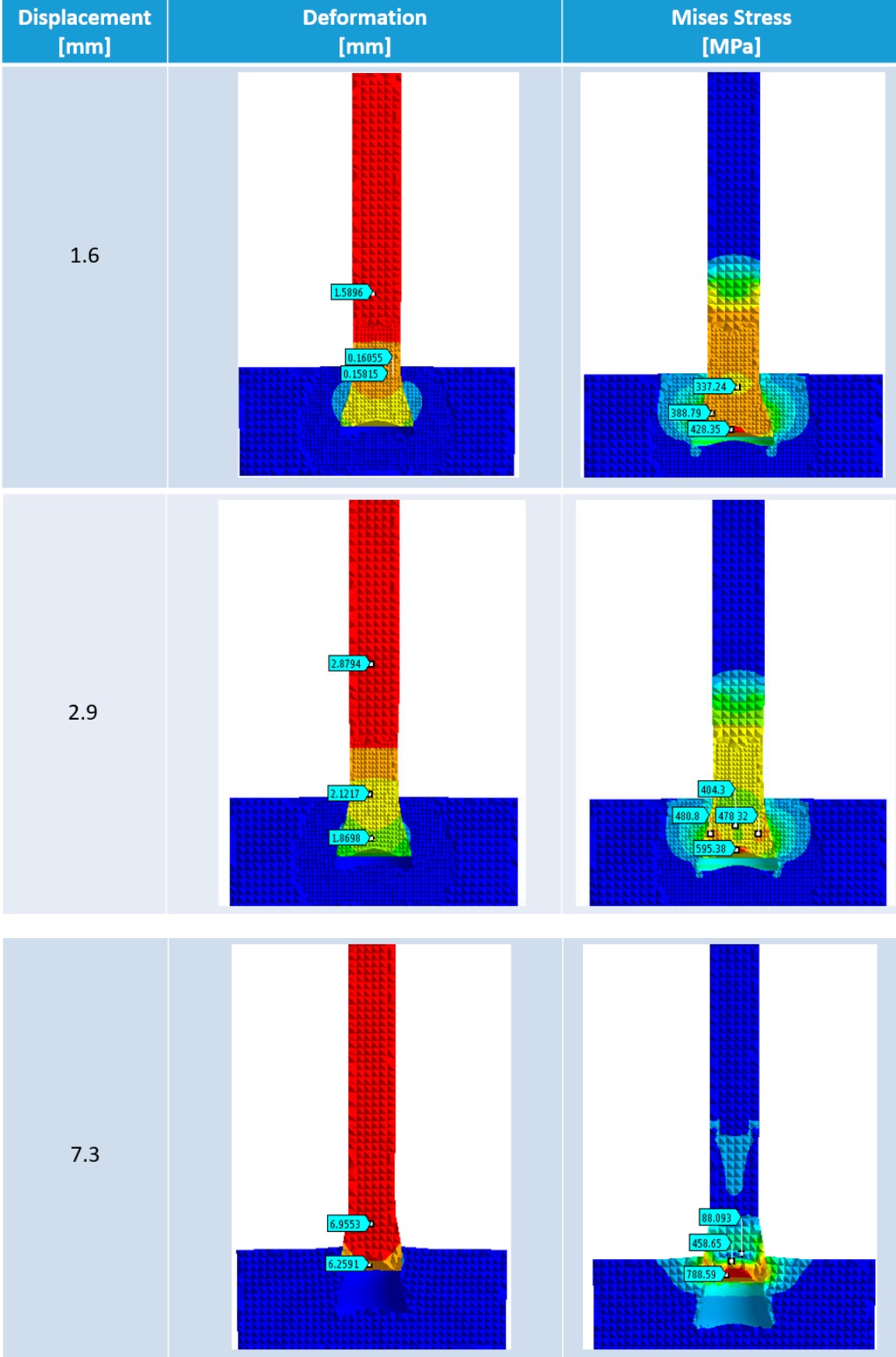

**Figure 7.** FE model displaying the development of Misses stresses of the friction riveted joints under tensile loading.

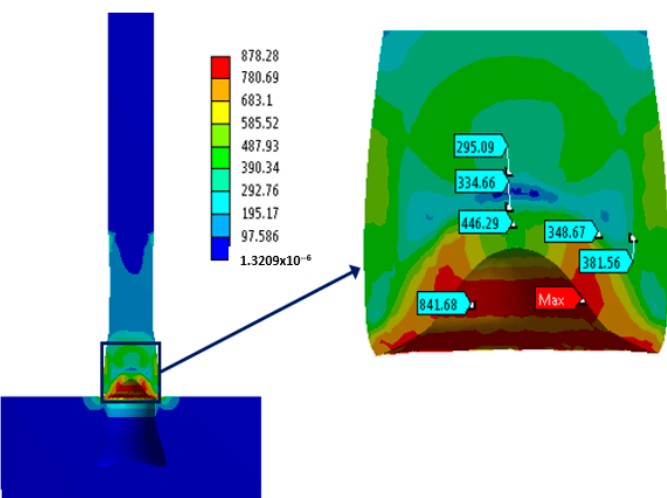

**Figure 8.** FE model of joint failure.

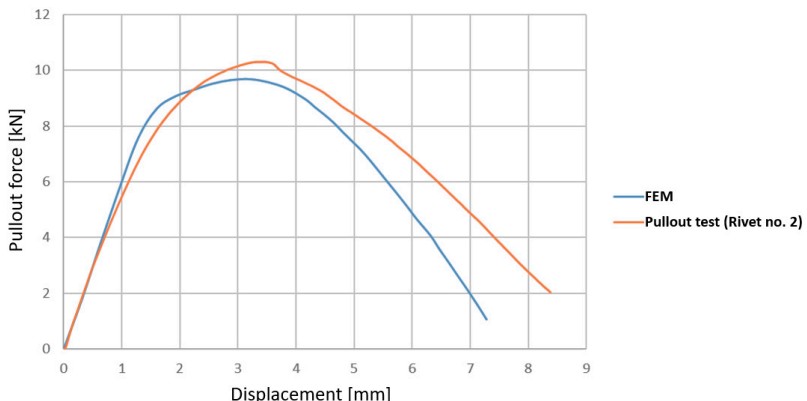

**Figure 9.** Comparison between experimental values and FE model of pullout tests.

A limited number of researchers have successfully attempted Friction Riveting. Hynes et al. performed low-speed Friction Riveting of AA1100 threaded rivets with polymethyl methacrylate (PMMA) using in-house adapted Friction Riveting equipment with a spindle capable of up to 3000 rpm [11]. The authors also reported the Type III failure for some of their joints, with SEM observations linking this to the rupture of the polymeric material due to the applied tensile load, similar to the current work. This was indicated by serrations on the surface of the molten PMMA [11]. Future research will investigate changes in physical-chemical properties within the PEI-AA2024 interface as well as the expelled flash material. Furthermore, the influence of threads on both joint formation and mechanical performance will have to be analyzed and correlated with the microstructural changes within the materials. Gagliardi et al. used an adapted milling machine for Friction Riveting of titanium grade two-holed cylindrical rivets with both pure and GF-reinforced polyamide 6 (PA6), showing that the spindle speed (referred as rotational speed in other publications on Friction Riveting) is the main process parameter affecting the mechanical performance/anchoring force [12].

## 5. Conclusions

The present paper demonstrated the transferability of Friction Riveting from dedicated laboratory equipment to modified simple drilling machines using a commercially available off-the-shelf motor for achieving relatively high rotational speeds and thus sufficient energy for rivet tip deformation and anchoring. This investigation shows that Friction Riveting can be performed with easy-to-achieve modification of low-cost machinery. FE models were in agreement with the experimentally obtained data, with rivet failure at 9500 ± 900 N

by full-rivet pullout. This failure type was previously reported for Friction Riveting as a rather weak joint and is related to the ductile character of the material or insufficient rivet tip deformation/anchoring. Further work on the optimization and characterization of the rivet–polymer interface will clarify the nature of the failure and the possibilities for strength increase. Increasing the heat input might improve the deformation and anchoring of the rivet. This could be achieved by increasing the joining force while keeping the relatively low rotational speed of the current study, as compared with previous published work. To the best knowledge of the authors, this is the largest rivet diameter (Ø 6 mm) successfully reported in Friction Riveting.

**Author Contributions:** The authors contributed to this research work as follows: conceptualization, L.B. and A.F.; methodology, L.B. and M.B.; modelling, A.B.; experiments and validation, A.F. and M.B.; resources, A.F.; data curation, all equally; writing—original draft preparation, A.B.; writing—review and editing, L.B.; supervision, A.F.; project administration, L.B. and A.F.; funding acquisition, A.F. All authors have read and agreed to the published version of the manuscript.

**Funding:** This research was funded by the Romanian Ministry for Research, Innovation and Digitalization through national grant number PN-III-P11.1-MCT-2018-0032.

**Institutional Review Board Statement:** Not applicable.

**Informed Consent Statement:** Not applicable.

**Data Availability Statement:** The data presented in this study are available on request from the corresponding author.

**Conflicts of Interest:** The authors declare no conflict of interest. The funders had no role in the design of the study; in the collection, analyses, or interpretation of data; in the writing of the manuscript; or in the decision to publish the results.

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
