# Peer review of "Process Transferability of Friction Riveting of AA2024-T351/Polyetherimide (PEI) Joints Using Hand-Driven, Low-Cost Drilling Equipment"

_processes, doi:10.3390/pr9081376_

Round 1

Reviewer 1 Report

The present study is certainly providing a new point of view on ' Friction Riveting'. This investigation promotes the transfer of friction riveting technology from laboratory equipment to self-made low-cost equipment and plays a positive role in cost control and application. However, there are some issues which should be improved and I must consider rejecting the article for publication:

  • The article should explain the practical application scenarios and service environment of plastic friction riveting in detail. Whether the pull-out test can effectively simulate the service environment, and whether the pull-out force can explain the connection effect of riveting.
  • A large amount of the paper is a summary of the work of others, and the design representation of the experiment is not sufficient. The experimental parameters are too single.
  • In the study of friction riveting, the rivet failed due to the pull out of full rivet at 9500N±900 N, but the reason was not deeply analyzed, and improvement measures were not put forward.

Author Response

Respected Reviewer,

Thank you for taking the time to revise our paper, which was a collaborative work between our two institutes. This was an opportunity to replicate the previous findings in works at the Helmholtz Institute Hereon (formerly Helmholtz-Zentrum Geesthacht, HZG) using machinery and equipment usually available in workshops and factories around the world, without significant investment and only simple modifications. This was thought to be an important step to facilitate the transfer of the technology towards applications in several industrial areas by using machinery already available.

Moreover, we used the opportunity of this collaboration and the modelling skills of our partners for approaching a more detailed description of the failure mode in Friction Riveting, based on the experimental results generated, which was indeed one of the previously reported weaker joint, namely full-rivet pullout.

We added further details in our manuscript, in order to address your points raised, as follows:

  • Application scenarios: lines 49-54
  • Parameters being too single: lines 120-122, whereas the condition tested and modelled was the only one for which joints were successfully produced
  • The reason for the behavior is explained in lines 161-164 and lines 219-223, with the optimization not being the scope of the paper but the knowledge generated in previous works on friction riveting gives us the assurance that increasing the heat input, either by rotational speed or by joining force, would improve the mechanical anchoring/deformation. Nonetheless, this has to be treated with care, as a larger rivet tip deformation might also reduce the amount of resistant polymer above the anchoring zone and thus induce a premature failure.

Kind regards,

Lucian Blaga, in the name of all co-authors.

Reviewer 2 Report

  • Lack of important information in Abstract, such as research motivation, quantitative research results.
  • Less literatures review and summary of related fields in introduction.
  • The core problem to be solved is not reflected in the research background.
  • In the part of results and discussion, besides the general description of the experimental phenomena, there should be subjective findings from the authors.
  • The study seems to have failed to produce effective results, the conclusion does not tell readers the contribution and value of the research based on the research work.

Author Response

Respected Reviewer,

Thank you for taking the time to revise our paper, which was a collaborative work between our two institutes. This was an opportunity to replicate the previous findings in works at the Helmholtz Institute Hereon (formerly Helmholtz-Zentrum Geesthacht, HZG) using machinery and equipment usually available in workshops and factories around the world, without significant investment and only simple modifications. This was thought to be an important step to facilitate the transfer of the technology towards applications in several industrial areas by using machinery already available.

Moreover, we used the opportunity of this collaboration and the modelling skills of our partners for approaching a more detailed description of the failure mode in Friction Riveting, based on the experimental results generated, which was indeed one of the previously reported weaker joint, namely full-rivet pullout.

We added further details in our manuscript, in order to address your points raised, as follows:

  • Abstract has been amended
  • Description of the core problems to be solved: lines 54-59 and 86-92
  • Findings from the authors: lines 168-177
  • Conclusion has been amended: lines 219-223

Kind regards,

Lucian Blaga, in the name of all co-authors.

Round 2

Reviewer 1 Report

This paper lacks an analysis of scientific issues and is more like an engineering report. In the paper, there is no classification study on the factors that affect the joint formation, and the failure analysis is not thorough enough. In addition, lines 39 to 48 and lines 77 to 86 are completely repeated in the text. In the tensile test, four specimens were mechanically tested  and failed by full rivet pullout, but in Figure 9, only the  displacement-stress curve of No. 2 sample is shown, which is not comprehensive.

Author Response

Dear reviewers,

We are thankful for your time and ideas for improving our manuscript and we hope to have addressed your points correctly.

  • Lines 39-48 and 77-86 were mistakenly repeated within the previous updating of the manuscript. Fault has been fixed now.
  • Figure 9 shows indeed an example of force-displacement curve for the tensile tests, which were all similar in behavior. The failed samples are shown all in Figure 6, with condition No. 2 being the second one from the left.
  • It was already shown for this material combination, that the joining force is the most significant process parameter for joint formation. Nonetheless, this was investigated until now only for 5 mm diameter rivets and using a specially designed laboratory equipment for Friction Riveting, with much higher rotational speeds capacity, among other (such as controlling). What we showed in this work is that one can go as low as 10.000 rpm in order to still produce high-speed friction riveted joints, with more optimization required in terms of mechanical performance. We provided further interpretation on this aspects in Lines 178-193.

Kind regards

L. Blaga

Reviewer 2 Report

Accept

Author Response

Dear reviewers,

We are thankful for your time and ideas for improving our manuscript and we hope to have addressed your points correctly.

Kind regards,

L. Blaga
